# Gut-derived peptide hormone receptor expression in the developing mouse hypothalamus

Lídia Cantacorps[1,2], Bethany M. Coull[1], Joanne Falck[1], Katrin Ritter[1], Rachel N. Lippert[1,2,3]*

1 Department of Neurocircuit Development and Function, German Institute of Human Nutrition Potsdam-Rehbruecke, Nuthetal, Germany, 2 German Center for Diabetes Research (DZD), München-Neuherberg, Germany, 3 NeuroCure Cluster of Excellence, Charité-Universitätsmedizin Berlin, Berlin, Germany

☯ These authors contributed equally to this work.
* Rachel.lippert@dife.de

**Data Availability Statement:** All relevant data are within the manuscript and its Supporting Information files.

**Funding:** This work was supported financially by the Deutsche Forschungsgemeinschaft (DFG,

## Abstract

### Objective

In adult organisms, a number of receptors have been identified which modulate metabolic processes related to peptides derived from the intestinal tract. These receptors play significant roles in glucose homeostasis, food intake and energy balance. Here we assess these classical metabolic receptors and their expression as well as their potential role in early development of hypothalamic neuronal circuits.

### Methods

Chow-fed C57BL6/N female mice were mated and hypothalamic tissue was collected from offspring across postnatal development (postnatal day 7–21). Subsequent qPCR and Western Blot analyses were used to determine mRNA and protein changes in gut-derived peptide hormone receptors. Correlations to body weight, blood glucose and circulating leptin levels were analyzed.

### Results

We describe the gene expression and dynamic protein regulation of key gut-derived peptide hormone receptors in the early postnatal period of the mouse brain. Specifically, we show changes to Gastric inhibitory polypeptide receptor (GIPR), glucagon-like peptide 1 receptor (GLP1R), and cholecystokinin receptor 2 (CCK2R) in the developing hypothalamus. The changes to GIPR and InsR seem to be strongly negatively correlated with body weight.

### Conclusions

This comprehensive analysis underscores the need to understand the roles of maternal-derived circulating gut hormones and their direct effect on offspring brain development.

German Research Foundation) under Germany´s Excellence Strategy – EXC-2049 – 390688087 (NeuroCure to RNL) and by the German Center for Diabetes Research (82DZD03D2Y and 82DZD03D03 to RNL). Additional funds were acquired from the Deutsche Forschungsgemeinschaft (DFG, German Research Foundation) – 491394008. The funders had no role in study design, data collection and analysis, decision to publish, or preparation of the manuscript.

**Competing interests:** The authors have declared that no competing interests exist.

**Abbreviations:** ARH, Arcuate nucleus of the hypothalamus; CCK, Cholecystokinin; CCK2R, Cholecystokinin receptor 2; DPP, Dipeptidyl peptidase; GIP, Gastric inhibitory peptide; GIPR, Gastric inhibitory polypeptide receptor; GDM, Gestational diabetes mellitus; GLP1, Glucagon-like peptide 1; GLP1R, Glucagon-like peptide 1 receptor; GPCR, G-protein coupled receptor; ISH, In situ hybridization; InsR, Insulin receptor; P, Postnatal day.

## 1. Introduction

A number of metabolic diseases in adult organisms are characterized by changes in circulating hormones and disruption to their receptor signaling pathways. This has classically been studied with the role of the peripherally-derived hormones insulin and leptin. More recently, a deeper understanding of gut-derived peptides and their function in the adult brain has raised interest in the role of these receptors and their potential modulation for the treatment of metabolic disease (reviewed in [1] and [2]). Many novel therapeutics with extreme success in treating metabolic diseases are analogues of gut-derived peptide hormones, such as semaglutide, a glucagon-like peptide 1 (GLP1) receptor (GLP1R) agonist [3, 4]. These pathways have been extensively studied in the adult organism. Indeed, many targeted therapeutics are aimed at these metabolic receptors, and are either widely used in patients or in preclinical phases [5]. However, a subpopulation of people being treated for metabolic disorders represent an increasing number of those suffering from obesity. Many of these people are also entering the pregnancy period, as 1 in 5 people are classified as obese prior to pregnancy [6]. The impact of pharmacological interventions used to treat obesity on the developing brain are not well studied, and therefore many are suggested to be discontinued in pregnancy or closely monitored. In the context of perinatal development, the metabolic environment of the mother can significantly influence the formation of the brain, and more specifically the hypothalamus in humans and animal models. In addition, a number of genes playing a role in this primary development of neuronal circuits have been assessed for their deficits in the adult brain, when their expression is modulated and therefore generates disease states. However, the dynamic regulation of these components in the early postnatal phase has never been studied and is relevant to understand the timing of neuronal circuit development and periods where metabolic or environmental perturbations may be most detrimental to the developing brain, especially within the hypothalamus.

In early development, critical roles for the classical metabolic hormone receptors insulin and leptin have been shown to play a role in the development of brain circuits, with a specific focus on the hypothalamus [7–9]. However, with increasing studies on gut-derived metabolic peptides leading to profound success in further deciphering the neuronal circuits involved in food intake and regulation of metabolism, it is critical to understand how these peptide hormones may also be affecting the developing brain. Nevertheless, to date, the expression of key gut-derived peptide hormone receptors in the developing brain has not been fully investigated. Of specific interest to this study is the expression of the Cholecystokinin (CCK) receptor 2 (CCK2R), Gastric inhibitory polypeptide receptor (GIPR) and GLP-1R. CCK2R responds to both forms of CCK, mainly secreted by the enteroendocrine I cells in the upper intestine, as well as gastrin, secreted by the G cells located in the stomach and the duodenum [10]. CCK binding to the CCK2R is known to affect glutamate release, resulting in attenuation of food intake. The CCK2R is highly expressed within the brain, specifically throughout the hypothalamus [11–13]. Gastric inhibitory peptide (GIP), also referred to as glucose-dependent insulinotropic polypeptide, is secreted from the K cells in the small intestine and duodenum, while GLP-1 is released from preproglucagon neurons in the brain [14] as well as L cells located in the distal portion of the small intestine and has been extensively studied for its effects in the brain of adult animals. In the periphery, both GIP and GLP1 increase insulin levels in response to glucose intake, whereas in the brain they contribute to memory formation and have an anorectic effect. GIPR expression and activity in the brain has been reported, specifically within brain nuclei involved in the control of behaviors such as food seeking and meal patterning as well as nausea [15–17]. Similarly, GLP-1R-expressing cells are enriched in several important brain areas playing a central role in feeding behavior and energy homeostasis in mice, including the hypothalamus, bed nucleus of the stria terminalis, amygdala, lateral septum, central

midbrain and the hippocampus, as well as brainstem nuclei and vagus [18–20]. All of these peptide receptors have increased in popularity as potential therapeutic targets for obesity and diabetes treatment. Compound treatment with GLP1 and GIP modulators as well as CCK agonists are being targeted for their therapeutic potential in the treatment of obesity and diabetes [21, 22]. Interestingly, a subset of these compounds is not recommended for use during pregnancy due to known defects discovered in animal models [23], further confirming a role for the signaling via these receptors during early developmental processes.

CCK as well as Gastrin, GLP-1 and GIP are all known to be affected during pregnancy and lactation. CCK itself is required for the establishment of maternal behavior and gastrin, which can also bind and act via the CCK2R, is known to fluctuate across human pregnancy, further indicating a potential role of this hormone in the developmental processes [24]. In addition, CCK transcript expression in the arcuate nucleus of the hypothalamus (ARH) is downregulated in lactating rodents [25]. However, more interesting fluctuations in these receptors and their counterpart hormones have been identified in states of metabolic dysfunction. During pregnancy, postprandial GLP1 levels are generally reduced, with a further pronounced effect in conditions of gestational diabetes mellitus (GDM) [26]. Furthermore, individuals born to mothers with GDM have lowered fasting levels of GLP-1 in adulthood [27]. In rodents, plasma GIP concentrations are elevated during lactation and neonates show increased circulating GIP levels throughout the suckling period [28]. In humans, circulating GIP levels are also altered in GDM [26].

Here, we identify key gut-derived peptide hormone receptor expression in the developing mouse hypothalamus in a sex-specific manner. With this insight we probed the regulation of the expression of these receptors throughout development in the mouse brain via mRNA as well as protein expression in both male and female offspring. We report differing expression profiles across postnatal development, increasing the complexity of these receptors and their role in the proper formation of neuronal circuits.

## 2. Results

### 2.1 Body weight, blood glucose and hypothalamic insulin receptor expression across early development

In order to assess the expression across development, hypothalamic tissue was collected at various timepoints throughout the postnatal period (Fig 1A). As expected, body weight increased significantly across this time period with no significant difference in body weight between the sexes noted [$F_{age}(3,45) = 103.7$, p<0.001]. However, a tendency for females to weigh less at weaning was present. In addition, blood glucose did not differ between sexes, but a significant effect of age was present [$F_{age}(3,45) = 5.535$, p = 0.0027]. Our analysis of insulin receptor (InsR) protein expression confirmed previous reports of reduced expression of InsR with increasing postnatal age in rodents [29, 30] as well as previous radioligand binding assays showing changes in insulin binding with age in different brain regions, including the hypothalamus [31]. Furthermore, our Western blot analysis of both sexes shows a significant effect of age, with the lowest InsR expression at weaning, as well as slightly different expression profile changes between the sexes, with females showing a significant decrease in expression between the second and third postnatal week [postnatal day (P) 13 vs P21, p = 0.0171].

### 2.2 Cholecystokinin receptor expression is sexually dimorphic across development

Quantitative PCR analysis of multiple timepoints throughout development show a significant overall effect of age on the expression of CCK2R mRNA in females (Fig 2D; W(DFn, DFd) =

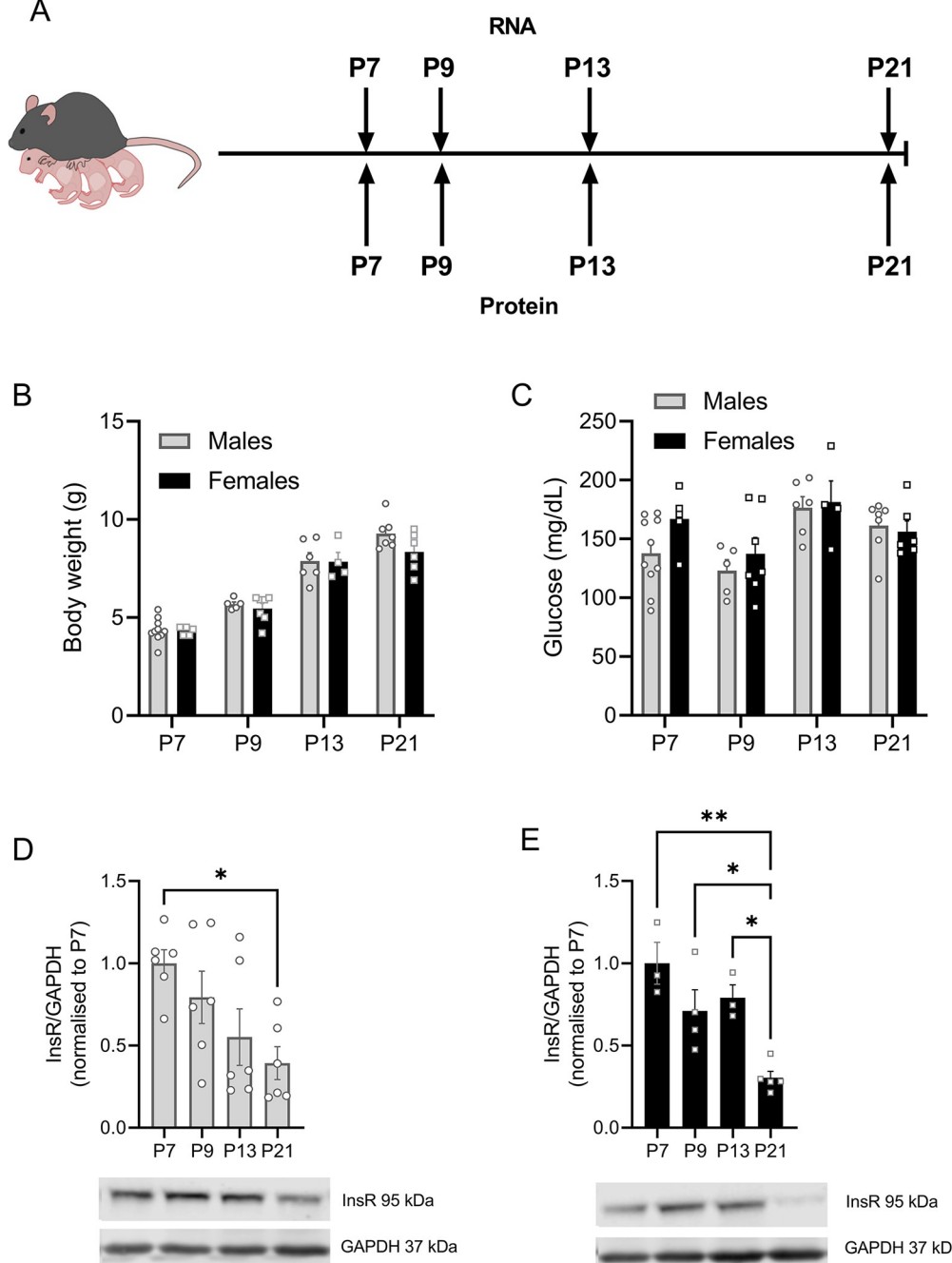

**Fig 1. Physiological characteristics across development and insulin receptor expression postnatally.** A) Hypothalamic samples were collected at timepoints across development for analysis. B) Body weight and C) blood glucose levels of male and female offspring were analyzed. InsR protein expression in D) males and E) females across early postnatal development. Open circles (males) or squares (females) represent individual data points. Data is plotted as Mean ± SEM * = p<0.05, ** = p<0.01.

5.912 (3.00, 7.973), p = 0.020), although no significant changes were found in males (Fig 2A; W(DFn, DFd) = 0.3945 (3.00, 10.38), p = 0.7597). Analysis of protein expression of the CCK2R depicts a much less variable amount of protein expression. Fig 2G shows a significant

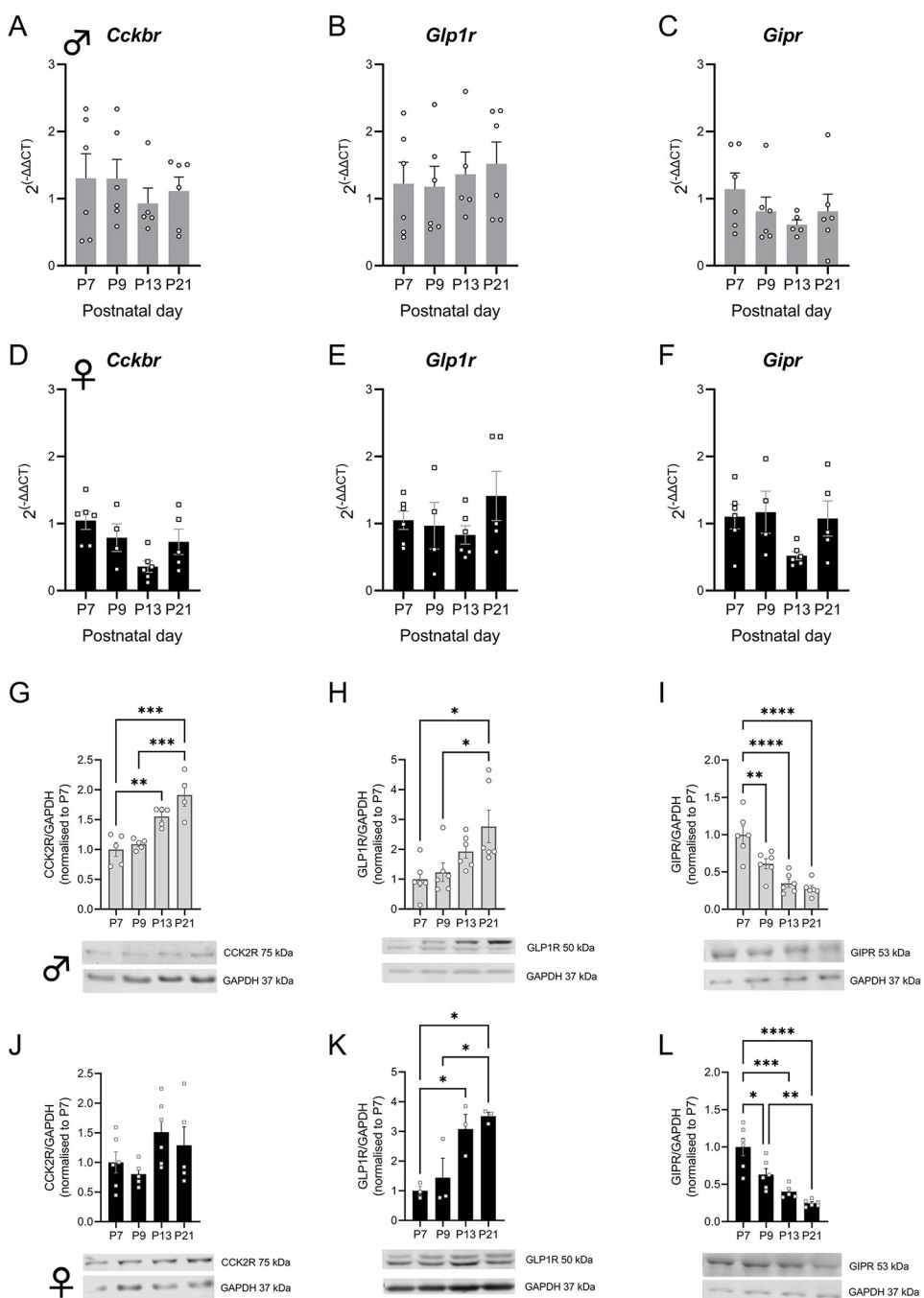

**Fig 2. Cholecystokinin 2 receptor, glucagon-like peptide receptor and Gastric inhibitory polypeptide receptor expression in postnatal development.** *Cckbr* mRNA expression across postnatal days in A) males and D) female animals. CCK2R protein expression in G) males and J) females across early postnatal development. *Glp1r* mRNA expression across postnatal days in B) males and E) female animals. GLP1R protein expression in H) males and K) females across early postnatal development. *Gipr* mRNA expression across postnatal days in C) males and F) female animals. GIPR protein expression in I) males and L) females across early postnatal development. Open circles (males) or squares (females) represent individual data points. Representative blots are shown for each protein assessed. Data is plotted as Mean ± SEM * = p<0.05, ** = p<0.01, *** = p<0.001, **** = p<0.0001.

increase in expression across postnatal days in male animals (F-statistic = 14.77, p<0.001). In females, no significant effect of time was noted (Fig 2J).

## 2.3 GLP1R expression in the hypothalamus increases across postnatal development in the hypothalamus

The GLP1R shows a unique pattern of gene and protein expression. In both sexes, mRNA expression did not significantly differ with age (Fig 2B and 2E; males: W(DFn, DFd) = 0.2096 (3.00, 10.42), p = 0.888; females: W(DFn, DFd) = 0.8020 (3.00, 7.842), p = 0.527). However, at the level of protein expression, both males and females showed a significant increase in GLP1R expression with increasing age (males: F-statistic = 4.950, p = 0.0099; females: F-statistic = 8.550, p = 0.0071, Fig 2I and 2L).

## 2.4 Hypothalamic GIPR expression decreases throughout development

Analysis of GIPR expression across development in the hypothalamus depicted an overall significant decline in mRNA expression across this time in female mice (Fig 2C, W(DFn, DFd) = 4.414 (3.00, 7.019), p = 0.0482), while no significant changes were observed in males (Fig 2F, W (DFn, DFd) = 1.491 (3.00, 9.462), p = 0.2791). Assessment of total GIPR protein levels in both males and female mice reflected the mRNA findings and depicted a highly significant decline in expression from P7 to P21 (Fig 2H and 2K, males p<0.0001, females p<0.0001).

## 2.5 Correlations of protein expression with glucose levels and body weight

To determine a potential role or link between receptor expression in the hypothalamus and functions in physiology in development, we analyzed correlations between postnatal blood glucose and body weight in males and females combined. No significant Pearson's correlations of INSR ($r^2$ = 0.00978, p = 0.566), CCK2R ($r^2$ = 0.09467, p = 0.0867), GLP1R ($r^2$ = 0.08392, p = 0.1020), or GIPR ($r^2$ = 0.07526, p = 0.0956) expression were noted with postnatal blood glucose levels (Fig 3A–3D). Interestingly, all four receptors were shown to be significantly correlated with body weight. GLP1R ($r^2$ = 0.4447, p<0.001) and CCK2R ($r^2$ = 0.4216, p<0.001) showed a positive correlation, increasing with increasing body weight (Fig 3A and 3C). However, INSR ($r^2$ = 0.4309, p<0.001) and GIPR ($r^2$ = 0.6738, p<0.001) showed significant negative correlations to postnatal weight (Fig 3B and 3D). GIPR hypothalamic expression also showed the least variable and more highly significant correlation to body weight.

## 2.6 Distribution of gene expression in the hypothalamus

In order to determine the likely most effected subregions of the hypothalamus due to the alterations across development, we targeted tissue collected to the individual hypothalamic subnuclei, including the ARC, VMH/DMH, PVH and LH from adult animals (S1 Fig). In adult mice, the distribution of the *Insr* expression is relatively evenly distributed throughout the hypothalamus, with the highest concentration of gene expression in the ARC and VMH/DMH (S1A Fig, [F(3, 21) = 8.787; p<0.001]). *Cckbr* appears to be widely distributed throughout the hypothalamus in both our data [F(3, 22) = 0.8842; p = n.s.] and in the *in situ* hybridization (ISH) data from Allen Brain Atlas (S1B Fig, [30]). *Glp1r* appears to be mainly expressed in the ARC, and to a lesser extent in the other subnuclei [F(3, 19) = 6.127; p = 0.004], which can also be seen on the ISH images from the Allen Brain Atlas (S1C Fig, [30]). *Gipr* is predominantly found in the ARC with expression also being identified in the LH, the VMH/DMH and the PVH [F(3, 22) = 4.778; p = 0.010]. To further probe the spatial distribution of gene expression in the early developmental period (Fig 4), we aligned ISH data from the Allen Developing

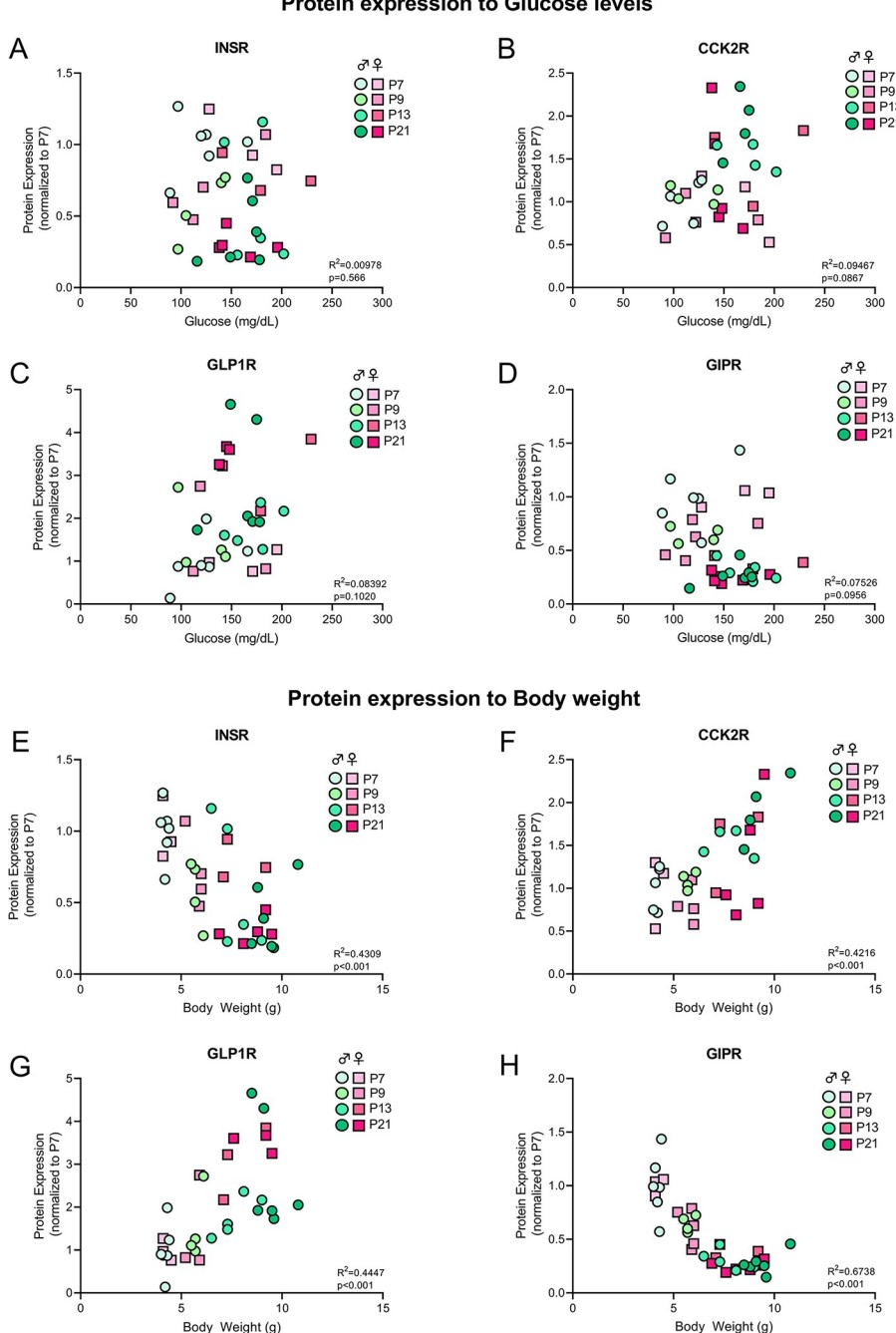

**Fig 3. Protein expression correlations with glucose levels and body weight.** INSR (A), CCK2R (B), GLP1R (C) and GIPR (D) protein expression across development in both sexes correlated with glucose (mg/dL) levels. INSR (E), CCK2R (F), GLP1R (G) and GIPR (H) protein expression across development in both sexes correlated with body weight. Circles (males) or squares (females) represent individual data points, color-coded by age.

Mouse Brain Atlas to determine the expression correlates across development for the *Insr*, *Cckbr*, and *Glp1r* (*Gipr* data not available). Here interestingly, in contrast to the compartmentalized expression of *Cckbr* and *Glp1r*, the gene expression of the *Insr* appears to be evenly distributed throughout the hypothalamus and displays what appears to be a slight decrease

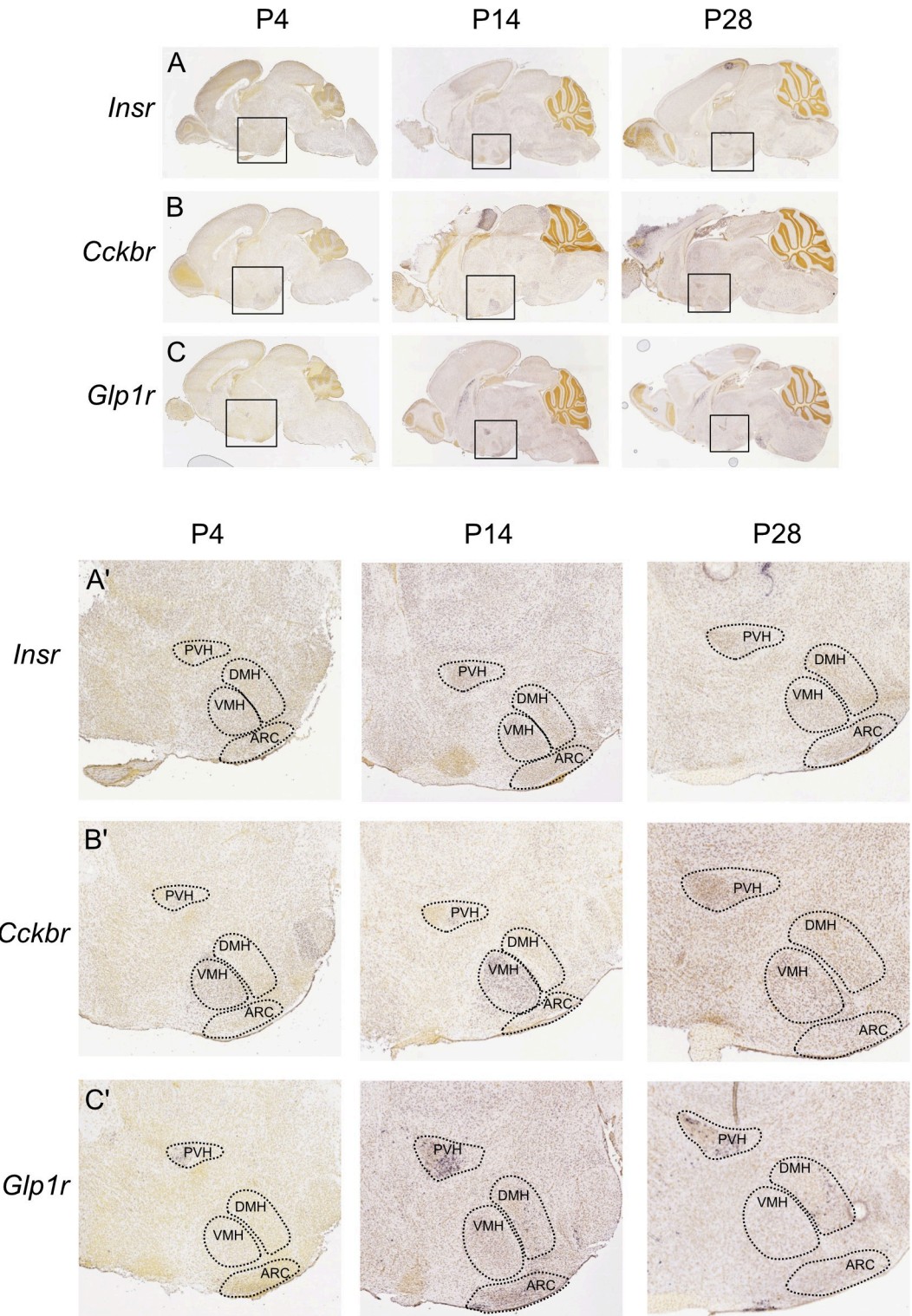

**Fig 4. Spatial distribution of *Insr*, *Cckbr* and *Glp1r* mRNA expression in the developing hypothalamus.** In situ hybridization images from the Allen Developing Mouse Brain Atlas (http://developingmouse.brain-map.org/) corresponding to the *Insr* (A, developingmouse.brain-map.org/gene/show/16110), *Cckbr* (B, developingmouse.brain-map.org/gene/show/12211) and *Glp1r* (C, developingmouse.brain-map.org/gene/show/14428) across early postnatal period (P4, P14 and P28) are represented.

between P14 and P28 (Fig 4A) which supports the Western blot data showing this same decrease in both males and females. In contrast to the wide expression of *Insr*, it can clearly be seen that the *Cckbr* is highly expressed in the VMH by P14(Fig 4B), despite being relatively dispersed in the hypothalamus of adult animals. Similarly, the *Glp1r* expression is highly compartmentalized into the PVH region in early development reaching its highest level by P14, while getting more widely distributed throughout the hypothalamic subnuclei with age (Fig 4C). Unfortunately, the only available developing brain ISH data is in male animals, warranting a further understanding as to the distribution within the hypothalamus subregions in female animals and their expression changes across development.

### 2.7 Protein expression changes across development and role of the postnatal leptin surge

In the early postnatal period it is known that particular hormonal surges are linked to postnatal growth and development, also within the brain. The hallmark of this is the postnatal leptin surge [32]. To determine a potential meditating effect of the postnatal leptin surge on expression of other key gut-derived peptide hormone receptors, we analyzed leptin levels in plasma collected from mice at P7, P9, P13, and P21. In both male and female animals, no correlation was noted between circulating leptin levels and protein expression of InsR ($r^2 = 0.0000345$, $p = 0.9750$), CCK2R ($r^2 = 0.03645$, $p = 0.3607$), GLP1R ($r^2 = 0.00155$, $p = 0.8392$) or GIPR ($r^2 = 0.001515$, $p = 0.8297$) (Fig 5A–5D). Thus, suggesting that inherent changes to leptin, known to affect growth and development of neuronal circuits in this early developmental time period, does not directly impact the expression of other gut-derived peptide hormone receptors.

## 3. Discussion

Growing evidence has shown that gut-derived peptides have both local and long-range effects throughout the whole organism. Specifically, these peptides binding to their subsequent receptors in the brain can modulate behaviors such as food intake and can change metabolic parameters. These peptides and their receptor signaling have been shown to change in states of obesity and altered metabolic health and have recently become targets for many pharmacological interventions. With the potential two-fold hit of obesity modulated receptor activity and expression profiles in combination with the known changes throughout pregnancy, an understanding of the function of these gut-derived metabolic factors in early development is critical. However, the role of these peptide receptors during critical periods of development have not been studied and potential implications for differences across sexes has not previously been presented. Here we show three major receptors for gut peptides and their regulation in the hypothalamus throughout postnatal periods, reporting the dynamic regulation of metabolic receptors across sex, age and body weight in early development.

Our results show that gut-derived peptide signaling during early postnatal development in mice, roughly equivalent to the third trimester in the context of brain development in humans [33], could indicate a multi-faceted role for these peptides in the formation of neuronal circuitry. Specifically, CCK and its action at CCK2Rs may prove to have different results in males and females as the expression profiles of protein levels differ between the sexes. In the context of GLP1R, an increasing protein expression across the postnatal period was observed in both sexes. In contrast, GIPR shows an opposite profile, a rapidly decreasing protein expression across development which suggests a role for this receptor and responsiveness to GIPR in the very early formation of hypothalamic neuronal networks. Interestingly, GIPR expression is also negatively correlated with body weight, which could reflect its involvement in the brain for neuronal outgrowth. The targeted analysis of the receptors' gene expression within the

## Protein expression to Leptin levels

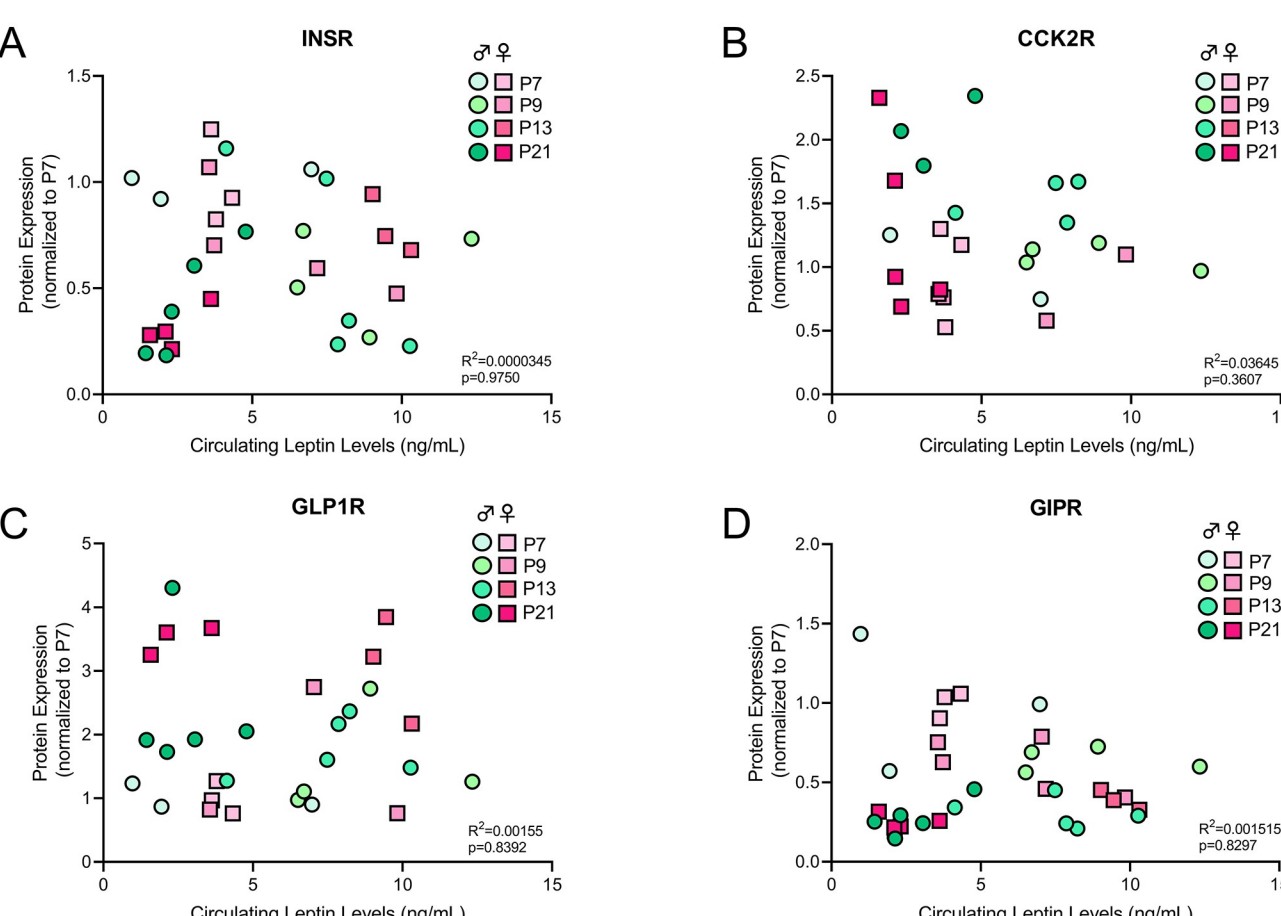

**Fig 5. Protein expression correlation with circulating leptin levels.** INSR (A), CCK2R (B), GLP1R (C) and GIPR (D) protein expression across development in both sexes was correlated with circulating leptin levels (ng/mL).

hypothalamic subnuclei in the adult mouse brain combined with the ISH data available from the Allen Developing Mouse Brain Atlas [30] allowed us to hypothesize which subregion in particular might be more affected by the protein expression changes here assessed during development. Potentially, the overall increased expression of CCK2R and GLP1R that we observed during early postnatal life could be compartmentalized into the VMH and the PVH, respectively. In contrast, the decline observed in both InsR and GIPR expression might be affecting the whole hypothalamus. Notwithstanding, a further analysis of the individual hypothalamic subnuclei expression profile across development would be necessary to corroborate this. While we aimed to dissect a link between gene and protein expression of the individual GPCRs, our data indicate that there is not a strong correlation between the two measurements. Indeed, while there is argumentation that mRNA and protein levels are strongly correlated [34, 35], these results are largely based on the analysis of highly abundant genes represented in most sequencing analysis. However, GPCRs are among the most abundant genes which are not detected in traditional sequencing analysis of mRNA abundance [36]. Indeed, among the top 50 most highly abundantly expressed GPCRs the lower limit is approximately 5 transcripts per million reads [37], and the transcripts we are detecting in this sample set are significantly lower in their expression, and thus more highly variable in their detection as well. GPCRs have

evolved to have very complex mechanisms of regulation, through desensitization, internalization and recycling, warranting their protein expression to be the more determinant factor in their role in development [38, 39]. Therefore, the findings of protein expression across early periods of life more strongly support a role of these proteins in mediating aspects of growth and development. Further studies will be necessary to uncover potential changes to their membrane expression and activity regulation in early development.

Despite the relative lack of studies in the literature focusing on the expression of these receptors on the developing hypothalamus, few studies have reported developmental changes in the brain of these gut-derived peptide receptors. Specifically, Nishimura *et al.* showed that *Cckbr* transcripts were not detected in the brain by *in situ* hybridization until late embryonic stages, but highly expressed in the neocortex by the second postnatal week [40]. Similar to our findings, increasing *Cckbr* expression in the primary somatosensory cortex of mouse brain over the first two postnatal weeks (P4, P6, P8, P10 and P14) was reported [30]. Furthermore, consistent with our study, decreasing expression of *Gipr* transcripts with increasing age has been reported in mouse cortical areas [41]. In addition, epigenetic mechanisms, such as DNA methylation, are known to play a key role in neurodevelopmental processes [42] and sex-dependent changes in the methylating enzymes have been reported in the hypothalamus during the first week of life [43]. Recently, MacKay *et al.* showed a differential methylation pattern of specific genes expressed in neurons of the ARH across postnatal development (P12-P35), including *Glp1r* and *Cck* in male mice, and *Gipr* and *Cck* in females, strengthening the role of these gut peptides on the hypothalamic development in a sex-specific manner [44].

In regard to the potential role of these gut peptides and their receptors on neuronal development, Teng *et al.* have recently identified GIPR as a potential promoter of neurite outgrowth [45] this is further supported by additional studies assessing growth of dorsal root ganglion neurons [46]. Furthermore, it has been shown that GIP has a neurotrophic action preventing neuronal cell death [47] and mice lacking GIPR show impaired hippocampal neurogenesis [48]. In fact, the use of a dual GIP/GLP-1 agonist enhanced hippocampal neurogenesis and reduced neuronal oxidative stress in mice [49]. Similarly, GLP-1/GLP1R signaling has been reported to play a neuroprotective role in experimental models of neurodegeneration. Enhancing GLP-1 availability by sitagliptin has been shown to increase axon regeneration and reduce neuronal apoptosis after spinal cord injury [50]. GLP1 and exendin-4, one of its long-acting analogs, have also been shown to have neurotrophic properties [51].

Incretins such as GLP-1 and GIP have been suggested to modulate maternal metabolism and fetal growth. Pregnant people show a reduced response to GLP-1, and this effect is even more evident in those with GDM. Postprandial GLP-1 and GIP levels are higher in GDM than in normoglycemic pregnancies [52], although in GDM basal levels are lower and reduced levels of these peptides may play a role in the dysregulation of glucose homeostasis post pregnancy [53]. Interestingly, lower levels of fasting GLP-1 in adult overweight offspring exposed to maternal diabetes have also been reported [27]. In rodents, maternal overnutrition during lactation reduced GLP-1R expression in the hypothalamus of male offspring, while higher CCK-R expression in the hypothalamus of females was present [54], suggesting that the gut-brain axis might be disrupted by early postnatal overfeeding. Further, alterations in GLP1R expression are also linked with disrupted postnatal development of pancreatic beta cells, compounding the potential effects on offspring glucose homeostasis capacity [55]. Strikingly, when the a GLP1 analog Exendin-4 was applied postnatally to mice, an almost complete loss of NPY immunoreactivity in the hypothalamus was seen for the life of the animal, further implicating the long-term negative consequences of disruptions to these gut derived peptides neuronal signaling pathways during early development [56].

GLP-1R agonists are a class of drugs targeting the incretin system that are now being investigated as potential medications to treat GDM, due to numerous recent publications showing their beneficial effects on insulin sensitivity and glucose intolerance during pregnancy [57–59]. However, the use of incretin drugs in pregnant people could also affect the developing fetus, highlighting the need for a better understanding of the action of these drugs during such a critical time period for offspring brain development. Likewise, dipeptidyl peptidase (DPP)-4 inhibitors, which prevent the degradation of endogenous GLP-1 and GIP [60], have been tested for GDM. Interestingly, maternal sitagliptin treatment, a highly selective DPP-4 inhibitor, has been shown to attenuate glucose intolerance and insulin resistance in male offspring rats at weaning [61]. Furthermore, dysregulated activity of the GLP-1 inactivating enzyme DPP-4 has been found in male mice from obese mothers, but when mothers were treated with sitagliptin, the progression of the obesity phenotype was delayed in males although no effects were observed in females [62]. Downregulated DPP-4 enzymatic activity in newborns of mothers with GDM compared to normoglycemic controls has also been demonstrated [63]. Altogether demonstrating that GLP-1/GLP-1R signaling cascades play a key role in the developmental programming of obesity, and thus effects of drugs targeting this system during pregnancy and lactation should be further examined.

Plasma CCK levels are reported to increase during pregnancy [64]. Interestingly, CCK and its receptor expression have been detected in both human and mouse placentas [65] and differential expression in the placenta of diabetic mothers has been found [66]. In the ARH, CCK expression was decreased during the course of lactation in rats [25], which could explain the hyperphagia accompanying this period. Furthermore, neonatal overnutrition induced a CCK-resistant phenotype in the offspring [67], while maternal malnutrition has been shown to reduce CCK levels in the offspring hypothalamus [68], which may contribute to overweight and metabolic dysfunctions.

Enhanced understanding of these peptide hormone receptors will likely implicate a clear developmental role of these factors, such as in the case of insulin and leptin, further underscoring that these factors which are usually studied for metabolic function in adults may prove to be critical for the timing of neuronal circuit formation in the developing brain. Thus, a clearer understanding for the role that these peptide hormones play in normal brain development needs to be in place prior to use of any peptide hormone analogues in sensitive developmental periods.

## 4. Material and methods

### 4.1 Animals

Wild-type C57BL6N (Charles River, Strain #027) mice were bred in on site facilities at the Max-Rubner Laboratory (MRL) at the German Institute for Human Nutrition Potsdam-Rehbruecke (DIfE). Male and female offspring were sacrificed at the postnatal days (P) 7, 9, 13 or 21 (Fig 1A). A minimum of 2–3 litters is represented in each timepoint. For assessing the distribution of the receptors' gene expression within the hypothalamic subnuclei, samples were collected from adult female mice aged 3–4 months old. Animals were group housed in individually vented cages (IVC) cages with *ad libitum* access to food and water with a 12on/12off light cycle and constant room temperature ($22 \pm 2°C$). All experiments were approved by the necessary local authorities and conducted in compliance with the ARRIVE guidelines and the EU directive 2010/63/EU.

### 4.2 Tissue collection

Mice were euthanized under anesthesia using inhaled isoflurane. Brain tissue was harvested and collection of the hypothalamus was performed using a 1 mm metal brain block. All tissue

was collected at the same time of day (ca. 16:00– 18:00 h) and stored at -70-80˚C until further processing. Blood glucose was measured directly from trunk blood using a handheld glucometer (Contour Care, Ascensia).

## 4.3 RNA extraction

Hypothalamus tissue was homogenized in 2 ml screw-cap tubes in 1ml in-house TRIzol reagent (Table 1) before adding 12.5 μl of glycogen (Genaxxon bioscience, Cat# M6015.0005) to each sample and vortexing thoroughly. 200 μl chloroform (Merck, Cat# 102445) was added to allow phase separation. Samples were vortexed, shaken vigorously for 15 seconds, and then incubated for 5 minutes at room temperature. Samples were centrifuged at 12,200 g for 10 minutes at 4˚C to yield 3 phases: an upper clear aqueous layer containing RNA, a white interphase containing DNA, and bottom pink layer containing protein. 500 μl of the upper aqueous layer was collected and added to the prepared tubes containing 650μl of ice-cold isopropanol (Carl Roth, Cat# 7343.1), Samples were inverted 2–3 times and then placed at -20˚C for 1 hour. Samples were centrifuged at 12,200 g for 15 minutes at 4˚C to collect the RNA pellet. After discarding the supernatant, 1 ml of 80% EtOH was added, and tubes were inverted 4–6 times to detach the pellet. Samples were centrifuged at 7,600 g for 15 minutes at 4˚C, supernatant was discarded and the pellets were air-dried for 7 minutes by leaving tubes under an extraction hood with the lids removed. Resuspension of the pellet was performed in 30 μl diethyl pyrocarbonate-treated water (DEPC) before adding 3.7 μl DNAse 10x Buffer (Fisher Scientific, Cat# B43), 2 μl Superasin (Fisher Scientific, Cat# AM2696), and 2 μl DNAse I (Fisher Scientific, Cat# EN0521) and mixing gently. Subsequent incubation of the samples at 37˚C took place for 25–30 minutes followed by thoroughly vortexing. Extracted RNA samples were stored at -70-80˚C until further use.

## 4.4 cDNA synthesis

Total RNA was measured using the Quantus Fluorometer instrument (Promega). cDNA reverse transcription of between 50 and 1000 ng of RNA from each sample was performed using the NZY first-strand cDNA synthesis kit (NZYtech, Cat# MB12502). Each PCR reaction (20μl final volume) was run at 50˚C for 30 min, 85˚C for 5 min, 37˚C for 20 minutes and then kept at 4˚C. cDNA was stored at -20˚C until further use.

## 4.5 qPCR

Quantitative RT-PCR was performed in a Quant Studio 12K Flex Real-Time PCR System. The primers (Table 2) used were preoptimized and validated. Amplifications were run in duplicates containing 10 ng cDNA diluted in nuclease-free water, 0.4 μM each of forward and reverse primers and 5 μl of NZY Speedy qPCR Green Master Mix (NZYtech, Cat# MB223) in a total volume of 10 μl. GAPDH was used as the housekeeping gene. The 2^-ΔΔCt comparative method was used to quantify the results which were normalized to the P7 time-point.

Table 1. Trizol composition.

| Compound | Concentration |
|---|---|
| Phenol | 38% |
| Guanidinium thiocyanate | 0.8 M |
| Ammonium thiocyanate | 0.4 M |
| Sodium acetate | 0.1 M |
| Glycerol | 5% |

**Table 2. Primer sequences used for qPCR.**

| Gene | Forward (5' -> 3') | Reverse (5' -> 3') |
|------|------|------|
| Cckbr | GATGGCTGCTACGTGCAACT | CGCACCACCCGCTTCTTAG |
| Gapdh | CGACTTCAACAGCAACTCCCACTCTTCC | TGGGTGGTCCAGGGTTTCTTACTCCTT |
| Gipr | CTCATCTTCATCCGCATCCT | GGAAACCCTGGAAGGAACTT |
| Glp1r | GGGTCTCTGGCTACATAAGGACAAC | AAGGATGGCTGAAGCGATGAC |
| Insr | AATGGCAACATCACACACTACC | CAGCCCTTTGAGACAATAATCC |

## 4.6 Protein extraction

Homogenization of tissue took place in 300 μL of lysis buffer [0.15 NaCl, 1% TX-100, 10% glycerol, 1 mM EDTA, 50 mM TRIS pH = 7.4] containing a phosphatase and protease inhibitor cocktail [Complete ULTRA Protease Inhibitor Cocktail Tablets (Cat# 5892970001) and PhosSTOP Inhibitor Cocktail Tablets (Cat# 4906837001) respectively; Roche]. Centrifugation of homogenates took place at 12,200 rpm for 20 min at 4°C and supernatants were collected for quantification of protein content using the Pierce BCA protein assay kit (ThermoFisher, Cat# 23225), following manufacturer's instructions.

## 4.7 Western blotting

For each sample equal amounts of total protein (12 μg) mixed with 5x loading buffer (153 mM TRIS pH = 6.8, 7.5% SDS, 40% glycerol, 5 mM EDTA, 12.5% 2-β-mercaptoethanol and 0.025% bromophenol blue) were loaded into 10% SDS-polyacrylamide gels, and transferred to a PVDF membrane (Immobilion-P, Millipore, Cat# IPFL00005). 5% BSA in Tris-buffered saline (TBS) (100 mmol/L NaCl, 10 mmol/L Tris, pH = 7.4) and 0.1% Tween-20 was used to block membranes for 1 hr and then immunoblotting of the membranes was performed overnight at 4°C by incubating with the primary antibodies listed in Table 3. Finally, a 1 hr incubation of the membranes with their respective secondary fluorescent antibodies: anti-mouse (1:2500, IRDye 800, Rockland, Cat# 610-132-121) and anti-rabbit (1:2500, IRDye 680, Rockland, Cat# 611-144-002) was performed. Immunoreactive bands were detected using a LI-COR Odyssey scanner (LI-COR Biosciences). Normalization of protein densities compared to the expression signal of the housekeeping gene in the same samples was performed and expressed relative to the P7 time-point.

## 4.8 ELISA

Trunk blood samples from pups aged 7, 9, 13 and 21 days old were collected in EDTA-tubes (Sarstedt; Cat#16,444), centrifuged at 2,500 g for 25 min at 4°C and plasma aliquots were stored at -80°C. Circulating leptin levels were quantified using the Mouse leptin ELISA kit (Crystal Chem; Cat# 90030, Lot# 21OCML444) following the manufacturer's instructions and absorbance was measured at 450/630 nm using a microplate reader (BMG labtech).

**Table 3. Primary antibodies used for Western blot.**

| Antigen | Host | Dilution | Supplier | Catalog no. |
|------|------|------|------|------|
| CCK2R | Mouse | 1:200 | Santa Cruz Biotechnology | sc-166690 |
| GAPDH | Mouse | 1:2500 | Santa Cruz Biotechnology | sc-365062 |
| GIPR | Rabbit | 1:1000 | Proteintech | 28322-1-AP |
| GLP1R | Rabbit | 1:1000 | Novus Biologicals | NBP1-97308 |
| InsR-β | Rabbit | 1:1000 | Cell Signaling | 3025 |

### 4.9 Statistical analysis

All data was analyzed using GraphPad Prism 9.2.0. For qPCR datasets with significant differences in variance within groups a Welch's ANOVA was used. Outliers in the qPCR analyses were determined using a Grubb's test for outliers, with an alpha = 0.01 and all data sets with outliers removed passed the Shapiro-Wilk Test for normality. Western blot data pertaining to changes across development were analyzed using a One-Way ANOVA and Bonferroni post hoc analysis for between timepoint comparisons. For data comparing between sexes, a Two-Way ANOVA was performed followed by Bonferroni post hoc tests. Results are plotted as mean ± standard error of the mean (SEM). Correlation analysis was performed using a Pearson correlation coefficient. The significance threshold was set at $p < 0.05$.

## Supporting information

**S1 Fig. Gene expression of *Insr*, *Cckbr*, *Glp1r* and *Gipr* across adult hypothalamic subnuclei.** In situ hybridization images from the Allen Mouse Brain Atlas (http://mouse.brain-map.org/) corresponding to *Insr* (A), *Cckbr* (B), *Glp1r* (C) and *Gipr* (D) transcripts expression across the adult hypothalamus were aligned with data obtained from adult female mice hypothalamic subnuclei dissection, including the arcuate nucleus (ARC), the ventromedial/dorsomedial nucleus of the hypothalamus (VMH/DMH), the paraventricular nucleus of the hypothalamus (PVH) and the lateral nucleus of the hypothalamus (LH). Expression of the *Insr* (mouse.brain-map.org/experiment/show/69735484), *Cckbr* (mouse.brain-map.org/experiment/show/69236993), *Glp1r* (mouse.brain-map.org/experiment/show/73606497) and *Gipr* (mouse.brain-map.org/experiment/show/70295936) in the adult mouse brain. Open circles represent individual data points. Data is plotted as Mean ± SEM.
(TIF)

**S1 File. Body weight, glucose levels, protein expression and leptin levels.** Raw data regarding body weight (g), blood glucose levels (mg/dl), relative protein expression normalised to P7 group and circulating leptin levels (ng/ml) for each animal used in this study are shown.
(XLSX)

**S2 File. qPCR data.** Raw qPCR data of Glp1r, Gipr, Cckbr and Gapdh across the 4 postnatal time-points in both males and females are shown, together with the analysis of their expression using the $2^{(-\Delta\Delta CT)}$ method.
(XLSX)

**S1 Raw images.**
(PDF)

## Acknowledgments

We would like to acknowledge Jiajie Zhu for contribution of hypothalamic tissue and Miguel Serrano for the extraction of RNA used in the manuscript. The image of mice from Fig 1 was sourced from SciDraw (https://scidraw.io/).

## Author Contributions

**Conceptualization:** Lídia Cantacorps, Bethany M. Coull, Joanne Falck, Rachel N. Lippert.

**Data curation:** Lídia Cantacorps, Bethany M. Coull, Joanne Falck, Katrin Ritter, Rachel N. Lippert.

**Formal analysis:** Lídia Cantacorps, Bethany M. Coull, Joanne Falck, Katrin Ritter, Rachel N. Lippert.

**Funding acquisition:** Rachel N. Lippert.

**Investigation:** Rachel N. Lippert.

**Methodology:** Lídia Cantacorps, Bethany M. Coull, Joanne Falck, Katrin Ritter, Rachel N. Lippert.

**Project administration:** Rachel N. Lippert.

**Resources:** Rachel N. Lippert.

**Software:** Rachel N. Lippert.

**Supervision:** Rachel N. Lippert.

**Validation:** Lídia Cantacorps.

**Visualization:** Lídia Cantacorps, Rachel N. Lippert.

**Writing – original draft:** Lídia Cantacorps, Rachel N. Lippert.

**Writing – review & editing:** Lídia Cantacorps, Bethany M. Coull, Joanne Falck, Rachel N. Lippert.

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
