## [Decision Letter · Decision Letter 0]

28 Apr 2023

PONE-D-23-05790Gut-derived peptide hormone receptor expression in the developing mouse hypothalamusPLOS ONE

Dear Dr. Lippert,

Thank you for submitting your manuscript to PLOS ONE. After careful consideration, we feel that it has merit but does not fully meet PLOS ONE’s publication criteria as it currently stands. Therefore, we invite you to submit a revised version of the manuscript that addresses the points raised during the review process.

 Your manuscript has been reviewed by a referee who is a recognized expert in this field. The reviewer recommended substantial revision. Please, carefully consider the reviewer's comments and revise your manuscript accordingly

We look forward to receiving your revised manuscript.

Kind regards,

Hubert Vaudry

Academic Editor

PLOS ONE

“This work was supported financially by the Deutsche Forschungsgemeinschaft (DFG, German Research Foundation) under Germany´s Excellence Strategy – EXC-2049 – 390688087 (NeuroCure to RNL) and by the German Center for Diabetes Research (82DZD03D2Y and 82DZD03D03 to RNL)”

“This work was supported financially by the Deutsche Forschungsgemeinschaft (DFG, German Research Foundation) under Germany´s Excellence Strategy – EXC-2049 – 390688087 (NeuroCure to RNL) and by the German Center for Diabetes Research (82DZD03D2Y and 82DZD03D03 to RNL). The funders had no role in study design, data collection and analysis, decision to publish, or preparation of the manuscript.”

Reviewers' comments:

Reviewer's Responses to Questions

**Comments to the Author**

1. Is the manuscript technically sound, and do the data support the conclusions?

Reviewer #1: Partly

2. Has the statistical analysis been performed appropriately and rigorously? 

Reviewer #1: No

3. Have the authors made all data underlying the findings in their manuscript fully available?

Reviewer #1: Yes

4. Is the manuscript presented in an intelligible fashion and written in standard English?

Reviewer #1: Yes

5. Review Comments to the Author

Reviewer #1: Cantacorps et al have examined gut-derived peptide receptor expression in the hypothalamus, at the gene and protein level, across the early post-natal period in both sexes. These results are of interest as the in utero environment can impact on hypothalamic development and lead to later loss of metabolic homeostasis and the development of obesity, but the mechanisms are unknown. In order to understand how hypothalamic development is altered, studies such as this one examining developmental processes under normal circumstances are first required. The introduction is clear and explains the relevance and importance of the study, and the methods are detailed and easy to follow. The discussion places the new results in the context of what is already known in this field, but would benefit for some text discussing the limitations of the findings in this study. Although the data is novel, overall the manuscript does not contain a lot of data. I have some other queries and concerns as detailed below:

- In Figure 1B and 1C there appears to be some clustering within timepoints of the body weight and glucose data. Is this related to litter (i.e. littermates sharing a body weight similar to each other, but not other animals?). Related to this point, the expression levels of the receptors at gene expression level is very varied within each time point, creating large error bars on the graphs. Is any of this explained by litter differences or is it correlated to body weight/ glucose?

- A limitation of this study is the authors have used whole hypothalamus which may mask changes happening in individual nuclei (relevant as these receptors are not expressed in all hypothalamic nuclei). I realise this will not be possible to correct post- experiment but do the authors have any pilot data on different nuclei? This limitation should be noted in the discussion.

- The authors have used the 2^-ΔΔCt comparative method to analyse qPCR results and state groups are normalised to the P3 timepoint. Is this within sex? (i.e. is it possible to compare between sexes to examine whether there are sex differences in mRNA quantity rather than just seeing the changes across a time course?). Also can the authors explain why Fig 2C the P3 time point is sitting at around 5 on the y axis- have the authors used a geomean?

- There also seem to be several individual points in the gene expression graphs in fig 2 where gene expression is nearly at zero. Can the authors explain this- do they think the gene expression is actually zero in these animals or is it a problem with tissue dissection/ RNA extraction/ reverse transcription? The spread of the data in each timepoint in Fig 2A-F is very variable aim this is hindering interpretation of the data.

- All of the genes examined seem to be at their lowest (in both males and females) around the P9-P13 timepoint. Can the authors confirm that this is not due to a technical error (where all the mice collected on the same day, or processed by a different researcher?). If this possibility is ruled out, have the authors considered an effect of the leptin surge which is occurring at this time point- do they have data on serum leptin level in the same mice to plot a correlation?

- There really is not much synergy between the gene and protein expression across the data. This could of course be due to post-transcriptional mechanisms, but this should be addressed in the discussion.

6. PLOS authors have the option to publish the peer review history of their article (what does this mean?). If published, this will include your full peer review and any attached files.

Reviewer #1: No

---

## [Author Response · Author response to Decision Letter 0]

6 Jul 2023

Reviewer 1: Thank you for the helpful comments and suggestions we feel they have helped us to significantly improve our submission. All specific responses to the comments are included in the response to reviewers document. 

Editors: Thank you for the opportunity to submit a revised manuscript. We hope to substantial improvements will result in positive response to our resubmission. Of note specifically, we have also removed the funder information from the MS and have kept the same statement used directly under the funder statement provided. All primary data, including Western blots, qPCR results and body weight and glucose data are now included in Supplemental Information.

---

## [Editor Report · Decision Letter 1]

1 Aug 2023

Gut-derived peptide hormone receptor expression in the developing mouse hypothalamus

PONE-D-23-05790R1

Dear Dr. Lippert,

We’re pleased to inform you that your manuscript has been judged scientifically suitable for publication and will be formally accepted for publication once it meets all outstanding technical requirements.

Kind regards,

Hubert Vaudry

Academic Editor

PLOS ONE
---

## [Editor Report · Acceptance letter]

9 Aug 2023

PONE-D-23-05790R1 

Gut-derived peptide hormone receptor expression in the developing mouse hypothalamus 

Dear Dr. Lippert:

I'm pleased to inform you that your manuscript has been deemed suitable for publication in PLOS ONE. Congratulations! Your manuscript is now with our production department. 

Kind regards, 

on behalf of

Dr. Hubert Vaudry 

Academic Editor

PLOS ONE